# Finite-Time Last-Iterate Convergence for Learning in Multi-Player Games

**Yang Cai**
Yale University
yang.cai@yale.edu

**Argyris Oikonomou**
Yale University
argyris.oikonomou@yale.edu

**Weiqiang Zheng**
Yale University
weiqiang.zheng@yale.edu

## Abstract

We study the question of last-iterate convergence rate of the *extragradient algorithm* by [Kor76] and the *optimistic gradient algorithm* by [Pop80] in multi-player games. We show that both algorithms with *constant step-size* have last-iterate convergence rate of $O(\frac{1}{\sqrt{T}})$ to a Nash equilibrium in terms of the gap function in smooth monotone games, where each player's action set is an *arbitrary convex set*. Previous results only study the unconstrained setting, where each player's action set is the entire Euclidean space. Our results address an open question raised in several recent works [HIMM19, GPD20, GPDO20], which ask for last-iterate convergence rate of either the extragradient or the optimistic gradient algorithm in the constrained setting. Our convergence rates for both algorithms are tight and match the lower bounds by [GPD20, GPDO20]. At the core of our results lies a new notion – the *tangent residual*, which we use to measure the proximity to a Nash equilibrium. We use the tangent residual (or a modification of the tangent residual) as the the potential function in our analysis of the extragradient algorithm (or the optimistic gradient algorithm).

## 1 Introduction

We consider learning in *monotone games*, a class of multi-player games introduced by [Ros65] that include many well studied games, e.g., two-player zero-sum games, convex-concave games, $\lambda$-cocoercive games [LZMJ20], zero-sum polymatrix games [BF87, DP09, CD11], and zero-sum socially-concave games [EDMN09]. We focus on the following question: *Can we obtain **last-iterate convergence rate** to a **Nash equilibrium** in monotone games when all players act according to a simple learning algorithm?*

We adopt the multi-player online learning model as introduced in [CBL06], where players interact with each other repeatedly. At every time step $t$, every player $i \in \{1, \dots, N\}$ chooses an action $z_t^{(i)}$ from her action set $\mathcal{Z}^{(i)}$, which we assume to be a closed convex set in $\mathbb{R}^{n_i}$. We say the game is *unconstrained* if $\mathcal{Z}^{(i)} = \mathbb{R}^{n_i}$ for each player $i$. Player $i$'s loss function $\ell_t^{(i)}(\cdot)$ is determined based on the underlying game and the actions of the other players in round $t$. Player $i$ receives the loss $\ell_t^{(i)}(z_t^{(i)})$ as well as some additional feedback that informs her how to improve her decisions in the future. In this paper, we assume that each player receives the gradient feedback, i.e., player $i$ receives the vector $\nabla \ell_t^{(i)}(z_t^{(i)})$. We make an additional mild assumption that the game is *smooth*, i.e., all players' gradients are Lipschitz. Smoothness is a natural assumption that is satisfied in most applications and is also made in the majority of works concerning monotone games.

36th Conference on Neural Information Processing Systems (NeurIPS 2022).

| Game class | Setting | Step size | Convergence rate |
|---|---|---|---|
| Strongly monotone | general | constant | $O(c^{-T})$ (see e.g., [Tse95] [LS19, MOP19, ZMM$^+$21]) |
| Cocoercive | unconstrained | constant | $O(\frac{1}{\sqrt{T}})$ [LZMJ20] |
| Monotone | general | constant | Asymptotic [Pop80, HIMM19] |
| | general | decreasing | Asymptotic $^*$ (see e.g., [ZMM$^+$17] [ZMA$^+$18, MZ19, HAM21]) |
| | unconstrained | constant | $O(\frac{1}{\sqrt{T}})^\dagger$ [GPD20] |
| | general | constant | $O(\frac{1}{\sqrt{T}})$ [**This paper**] |

Table 1: Last-iterate convergence for no-regret learning in smooth monotone games with perfect gradient feedback. (*) The results hold for variationally stable games. (†) The result holds under an additional second-order smoothness assumption.

The standard metric to quantify an online learning algorithm's performance is the regret. Formally, the regret of player $i$ is defined as the difference between $\sum_{t=1}^{T} \ell_t^{(i)}(z_t^{(i)})$, player $i$'s cumulative loss, and $\min_{z \in \mathcal{Z}^{(i)}} \sum_{t=1}^{T} \ell_t^{(i)}(z)$, the loss incurred by the best fixed action in hindsight. An online learning algorithm is no-regret if, even under an adversarially chosen loss sequence $\{\ell_t^{(i)}(\cdot)\}_{t \in [T]}$, its regret at the end of round $T$ is sublinear in $T$.

A vast literature on learning in games discusses the convergence to a Nash equilibrium using no-regret learning algorithms. However, most of the results concern only the *time-average* convergence, i.e., the convergence of the time average of the joint action profile, rather than the last-iterate convergence, i.e., the convergence of the joint action profile. From a game-theoretic perspective, the last-iterate convergence is more appealing compared to the time-average convergence, as only the last-iterate convergence provides a description of the evolution of the overall behavior of the players. In contrast, the trajectory of the players' joint action may be cycling around in the space perpetually while still converges in the time-average sense as demonstrated by [MPP18]. Recently, a line of work is devoted to obtain last-iterate convergence in smooth monotone games [ZMB$^+$17, ZMM$^+$17, ZMA$^+$18, DP18, MOP19, HIMM19, LNPW20, GPD20, LZMJ20, ZMM$^+$21]. However, unless the game is strongly monotone or unconstrained, only asymptotic convergence is known. Moreover, many of these results crucially rely on decreasing step-size, which, as pointed out by [LZMJ20], is unnatural from an economic point of view, because it treats newly acquired information with decreasing importance. Hence, the following question is of particular interest and is raised as an open question in [HIMM19, LZMJ20, GPD20].

> *Can we establish **last-iterate rates** if all players of a **constrained** smooth monotone game*     (*)
> *act according to a no-regret learning algorithm with **constant step size**?*

[LZMJ20] first realizes the importance of question (*) and takes initial steps towards addressing it. They show that if all players follow the *gradient descent* algorithm with constant step size, then for all smooth $\lambda$-cocoercive games, the joint action $(z_t^{(1)}, \ldots, z_t^{(N)})$ has last-iterate convergence rate of $O(\frac{1}{\sqrt{T}})$ to a Nash equilibrium in terms of the gap function. For smooth strongly monotone games, a subclass of $\lambda$-cocoercive games, linear last-iterate convergence rates are known [Tse95, Mal15, LS19, MOP19, ZMM$^+$21]. Despite the generality of $\lambda$-cocoercive games, several fundamental classes of games such as two-player zero-sum games, zero-sum polymatrix games [BF87, DGP09, CD11] and its generalization zero-sum socially-concave games [EDMN09] are monotone but not $\lambda$-cocoercive. [GPD20] extends the result to smooth monotone games with an $O(\frac{1}{\sqrt{T}})$ last-iterate convergence rate using a different algorithm – the *optimistic gradient* by [Pop80] under an additional second-order smoothness assumption. Note that gradient descent has been observed to fail to converge in even two-player zero-sum games (see e.g., [DISZ17]), so a different algorithm is indeed needed. However, the results by [LZMJ20] and [GPD20] only consider *unconstrained* games, while in most game-theoretic settings, the players' actions are constrained. For example, in finite games, a player is restricted to choose a distribution over her finite set of actions. We summarize the results for last-iterate convergence in monotone games in Table 1.

**Our Contributions.** Our first contribution is to provide an affirmative answer to question (*).

**Contribution 1:** In Theorem 3, we show that if all players of a constrained smooth monotone game act according to the optimistic gradient algorithm, which is no-regret, with a **constant step size**, then their joint action exhibits a last-iterate convergence rate of $O\left(\frac{1}{\sqrt{T}}\right)$ in terms of the gap function (Definition 2) to a Nash equilibrium.

Our result holds in the constrained setting and does not rely on the second-order smoothness assumption made in [GPD20]. Moreover, our upper bound is tight and matches the lower bound of [GPD20].

The problem of finding a Nash equilibrium in a smooth monotone game is essentially equivalent to solving a Lipschitz and monotone variational inequality (VI) (Definition 5 in Appendix D),[1] which has been studied since the 1960s [HS66, Bro65, LS67, BS68, Sib70]. There is a vast literature on solving VIs, and we refer the reader to [FP07] for further references. The extragradient (EG) algorithm by [Kor76] and the optimistic gradient (OG) algorithm by [Pop80] are arguably the two most classical and popular methods for solving monotone VIs. Despite the long history, the last-iterates of both algorithms are only known to asymptotically converge to a solution of the monotone and Lipschitz VI,[2] but no upper bounds on the rate of convergence had been provided for the general setting.

**Contribution 2:** We provide the first and tight last-iterate convergence rate of $O(\frac{1}{\sqrt{T}})$ in terms of the gap function for both EG and OG with constant step size for solving Lipschitz and monotone VIs (Theorem 9 and Theorem 10 in Appendix I).

Our analysis of OG for monotone games directly applies to monotone VIs. To be more consistent with our discussion of OG, we state our last-iterate convergence rate of EG in the context of online learning in monotone games (Theorem 3).[3] Prior to our work, last-iterate convergence rate of EG only exists in the unconstrained setting. [GPDO20] shows a $O(\frac{1}{\sqrt{T}})$ upper bound in terms of the gap function under an additional second-order smoothness condition. [GLG21] improves the result and shows that the same upper bound holds without the second-order smoothness condition but still requires the setting to be unconstrained.

**Our Analysis.** As mentioned in [GPD20], "The lack of existence of a natural potential function in general monotone games is a significant challenge in establishing last-iterate convergence." Indeed, many of the natural quantities such as the gap function, the norm of the gradient, the difference of two consecutive iterates are provably non-monotone even in bilinear games. See Appendix J for more discussion and examples. We propose a notion that (i) measures the proximity to a Nash equilibrium, and (ii) can be used to construct natural potential functions to analyze both EG and OG in monotone games. We call the new notion the **tangent residual**, which can be viewed as the norm of the gradient projected to the tangent cone of the current iterate (Definition 4). The tangent residual plays a crucial role in our analyses for both algorithms. Unlike the other quantities mentioned above, we show that the *tangent residual is monotonically decreasing* and has *a last-iterate convergence rate of $O(\frac{1}{\sqrt{T}})$* for EG. For OG, we prove that a small modification of the tangent residual is monotonically decreasing, which implies that the tangent residual has a last-iterate convergence rate of $O(\frac{1}{\sqrt{T}})$. Using the convergence rate of the tangent residual, we can easily derive the last-iterate convergence rate of other classical performance measures such as the gap function and the total gap function. However, we suspect these rates can be challenging to obtain directly without the help of a potential function. Finally, we establish the monotonicity of our potential functions using computer-aided proofs based on **sum-of-squares (SOS)**

---

[1] Technically, the definition of Lipschitz and monotone VI captures finding a Nash equilibrium in a smooth monotone game as a special case, but the difference has little impact on our analyses, and our results hold for Lipschitz and monotone VIs.

[2] The last-iterate asymptotic convergence of EG can be found in [Kor76] and [FP07], and the last-iterate asymptotic convergence of OG can be found in [Pop80] and [HIMM19].

[3] Although EG is not a no-regret learning algorithm (see Proposition 10 in [GPD20]), it is nevertheless a simple and natural learning algorithm.

**programming** [Nes00, Par00, Par03, Las01, Lau09]. Indeed, our potential function of OG is directly constructed using SOS programming. Additionally, our computer-aided proofs can be easily verified by humans. We think the tangent residual and the SOS-based analysis might be of independent interest. See Section 4.2, Appendix B, and Appendix C for more discussion.

## 1.1 Related Work

**Last-Iterate Convergence Rate for EG/OG like Algorithms.** [GPDO20, GPD20] show a lower bound of $\Omega(\frac{1}{\sqrt{T}})$ for solving bilinear games using any p-SCLI algorithms, which include EG and OG. In the unconstrained setting, if we further assume that either the game is strongly monotone or the payoff matrix $A$ in a bilinear game has all singular values bounded away from 0, linear convergence rate is known for EG, OG, and several of their variants [DISZ17, GBV$^+$18, LS19, MOP19, PDZC20, ZY19]. The results for the constrained setting are sparser. Unless the game is strongly monotone, most results only guarantee asymptotic convergence, i.e., converge in the limit, [DP18, LNPW20]. Finally, a recent paper by [WLZL21b] provides a linear convergence rate of OG for bilinear games when the domain is a polytope. They show that there is a *problem dependent* constant $0 < c < 1$ that depends on the payoff matrix of the game as well as the constraint set, so that the error shrinks by a $1 - c$ factor per iteration. However, $c$ may be arbitrarily close to 0, even if we assume the corresponding operator to be $L$-Lipschitz. Overall, their "problem-dependent" bound is incomparable and complements the worst-case view taken in this paper, where we want to derive the worst-case convergence rate for all smooth and monotone games. Our results are the first last-iterate convergence rates in this worst-case view and match the lower bounds by [GPDO20, GPD20].

**Other Algorithms and Performance Measures.** It is well-known that both EG [Nem04] and OG [HIMM19, MOP20] have a time-average convergence rate of $O(\frac{1}{T})$ in terms of the gap function for smooth monotone games. Other than the gap function, one can also measure the convergence using the norm of the operator if the setting is unconstrained, or the natural residual (Definition 7 in Appendix E) or similar notions if the setting is constrained. In the unconstrained setting, [Kim21], [YR21], and [LK21] provide algorithms that obtain $O(\frac{1}{T})$ convergence rate in terms of the norm of the operator, which is shown to be optimal by [YR21] for Lipschitz and monotone VIs. In the constrained setting, [Dia20] shows the same $O(\frac{1}{T})$ convergence rate under the extra assumption that the operator is cocoercive and loses an additional logarithmic factor when the operator is only monotone. Our result implies a $O(\frac{1}{\sqrt{T}})$ last-iterate convergence rate in terms of the natural residual or the gap function for both EG and OG. A main motivation of this paper is game-theoretic, that is, we would like to view *natural* learning algorithms as models of agents' behavior in online learning and understand the speed for the overall behavior to converge to a Nash equilibrium. Although some of the results above are natural for games, they either only hold in the unconstrained setting or do not converge in the last-iterate sense. From this game-theoretic view point, we believe understanding the last-iterate convergence rate of these natural learning algorithms is an important problem.

## 2 Preliminaries

We consider the Euclidean Space $(\mathbb{R}^n, \|\cdot\|)$, where $\|\cdot\|$ is the $\ell_2$ norm and $\langle\cdot,\cdot\rangle$ denotes inner product on $\mathbb{R}^n$. A **continuous game** $\mathcal{G}$ is denoted as $(\mathcal{N}, \{\mathcal{Z}^{(i)}\}_{i\in[N]}, \{f^{(i)}\}_{i\in[N]})$ where there are $N$ players $\mathcal{N} = \{1, \cdots, N\}$. Player $i \in \mathcal{N}$ chooses action from a closed convex set $\mathcal{Z}^{(i)} \subseteq \mathbb{R}^{n_i}$ such that $\mathcal{Z}_{\mathcal{G}} := \Pi_{i\in\mathcal{N}}\mathcal{Z}^{(i)} \subseteq \mathbb{R}^n$ and wants to minimize her cost function $f^{(i)} : \mathcal{Z}_{\mathcal{G}} \to \mathbb{R}$. For each player $i$, we denote by $z^{(-i)}$ the vector of actions of all the other players and by $z^{(i)}$ the action of player $i$. When players play according to action profile $z \in \mathcal{Z}_{\mathcal{G}}$, player $i$ receives gradient feedback $\nabla_{z^{(i)}} f^{(i)}(z^{(1)}, \ldots, z^{(N)})$. A *Nash equilibrium* of game $\mathcal{G}$ is an action profile $z^* \in \mathcal{Z}_{\mathcal{G}}$ such that $f^{(i)}(z^*) \leq f^{(i)}(z'^{(i)}, z^{*(-i)})$ for any $z'^{(i)} \in \mathcal{Z}^{(i)}$.

Let $F_{\mathcal{G}}(\cdot) = (\nabla_{z^{(1)}} f^{(1)}(\cdot), \cdots, \nabla_{z^{(N)}} f^{(N)}(\cdot))$ be an operator that maps any joint action in $\mathcal{Z}$ to the corresponding joint gradient feedback vector in $\mathbb{R}^n$. When $\mathcal{G}$ is clear from context, we omit the subscript and write $F_{\mathcal{G}}(z)$ ($\mathcal{Z}_{\mathcal{G}}$ resp.) as $F(z)$ ($\mathcal{Z}$ resp.).

**Nash Equilibria of Monotone Games.** Throughout this paper, we focus on smooth monotone games (Definition 1). It is well known that finding a Nash equilibrium of a monotone game is exactly the same as finding a solution to the variational inequality with monotone operator $F_{\mathcal{G}}$ (Lemma 1).

**Definition 1** ([Ros65]). *We say game $\mathcal{G}$ is L-smooth and* monotone *if the operator $F_{\mathcal{G}}$ is L-Lipschitz (i.e., $\forall z, z' \in \mathcal{Z}$, $L \cdot \|z - z'\| \geq \|F_{\mathcal{G}}(z) - F_{\mathcal{G}}(z')\|$) and monotone (i.e., $\forall z, z' \in \mathcal{Z}$, $\langle F_{\mathcal{G}}(z) - F_{\mathcal{G}}(z'), z - z' \rangle \geq 0$).*

**Remark 1** (Monotone and Concave Games). *For any monotone game $\mathcal{G} = (\mathcal{N}, \{\mathcal{Z}^{(i)}\}_{i \in \mathcal{N}}, \{f^{(i)}\}_{i \in [N]})$, the monotonicity condition implies that for any fixed $z^{(-i)}$, player i's cost function is convex in $z^{(i)}$. Thus, all monotone games are concave games. However, the converse is not true as illustrated in the following example.*

*Consider a two player game $\mathcal{G}$ where player 1 (or player 2) chooses action $x \in \mathbb{R}$ (or $y \in \mathbb{R}$), and their cost functions are $f^{(1)}(x, y) = f^{(2)}(x, y) = x \cdot y$. Clearly, $f^{(1)}(x, y)$ (or $f^{(2)}(x, y)$) is convex in $x$ (or $y$) if we fix $y$ (or $x$). It is not hard to see that $F_{\mathcal{G}}(x, y) = (y, x)$ for any $x, y \in \mathbb{R}$. Therefore, the game is not monotone as $\langle F_{\mathcal{G}}(x, y) - F_{\mathcal{G}}(y, x), (x, y) - (y, x) \rangle = -2(x - y)^2 < 0$ for any $x \neq y$.*

**Lemma 1** (1.4.2 Proposition [FP07] ). *For a monotone game $\mathcal{G}$, an action profile $z^*$ is a Nash equilibrium if and only if $\langle F_{\mathcal{G}}(z^*), z^* - z \rangle \leq 0, \forall z \in \mathcal{Z}$.*

**Remark 2.** *One sufficient condition for a Nash equilibrium $z^*$ to exist is when the set $\mathcal{Z}$ is bounded, but there are also other sufficient conditions that apply to unbounded $\mathcal{Z}$. See [FP07] for more details. Throughout this paper, we only consider monotone games that have a Nash equilibrium.*

**Definition 2** (Gap and Total Gap Function). *For a monotone game $\mathcal{G}$, two standard ways to measure the proximity of an action profile $z \in \mathcal{Z}$ to Nash equilibrium, are by its gap function and total gap function. Let $D$ be a fixed parameter. The gap function is defined as $\text{GAP}_{\mathcal{G},D}(z) = \max_{z' \in \mathcal{Z} \cap \mathcal{B}(z,D)} \langle F_{\mathcal{G}}(z), z - z' \rangle$, where $\mathcal{B}(z, D)$ is a ball with radius $D$ centered at $z$.[4] The total gap function is defined as $\text{TGAP}_{\mathcal{G},D}(z) = \sum_{i \in \mathcal{N}} (f^{(i)}(z) - \min_{z'^{(i)} \in \mathcal{Z}^{(i)} \cap \mathcal{B}(z^{(i)}, D)} f^{(i)}(z'^{(i)}, z^{(-i)}))$.*

*When $\mathcal{G}$ and $D$ are clear from context, we omit subscripts and write the gap function (total gap function resp.) at $z$ as $\text{GAP}(z)$ ($\text{TGAP}(z)$ resp.).*

**Remark 3.** *The total gap function is the sum of the suboptimality gaps, i.e., a player i's suboptimality gap is her cost under $z$ minus her cost after best responding to $z^{(-i)}$ within $\mathcal{B}(z^{(i)}, D)$. As the suboptimality gap is nonnegative for every player, a small total gap implies that no player can deviate from their current action to significantly improve their cost, and the action profile is close to a Nash equilibrium. Thus the total gap is $0$ if and only if players are at a Nash Equilibrium.*

*In Lemma 2, we show that the gap function with radius $D$ is an upper bound of the total gap with radius $\frac{D}{\sqrt{N}}$, where $N$ is the number of players, so if an action profile has small gap function, then again no player can deviate to significantly reduce their cost. A corollary of Lemma 1, is that the gap function is $0$ if and only if players are at a Nash Equilibrium.*

**The Optimistic Gradient Algorithm.** Let $w_k$ be the action profile played at day $k$ and assume that player $i$ updates her action according to the OG algorithm. For arbitrary $z_0^{(i)}$ and $w_0^{(i)}$, player's $i$ action at day $k + 1$ is $w_{k+1}^{(i)}$ such that $z_k^{(i)} = \Pi_{\mathcal{Z}^{(i)}} \left[ z_{k-1}^{(i)} - \eta \nabla_{z^{(i)}} f^{(i)}(w_k^{(i)}) \right]$ and $w_{k+1}^{(i)} = \Pi_{\mathcal{Z}^{(i)}} \left[ z_k^{(i)} - \eta \nabla_{z^{(i)}} f^{(i)}(w_k^{(i)}) \right]$.

---

[4]When $\mathcal{Z}$ is bounded, we choose $D$ to be the diameter of $\mathcal{Z}$. When $\mathcal{Z}$ is unbounded, the standard choice is to choose $D$ so that all the iterates of the algorithms are guaranteed to maintain in the ball $\mathcal{B}(z, D)$. See Appendix D for more details.

Note that action $z_k^{(i)}$ is not being played by player $i$ and is only used to compute action $w_{k+1}^{(i)}$. When all the players update their actions according to OG with step-size $\eta$, let $w_0 = (w_0^{(1)}, \ldots, w_0^{(N)})$ and $z_0 = (z_0^{(1)}, \ldots, z_0^{(N)})$. Then at day $k+1$, players pick action profile $w_{k+1}$, where:

$$z_k = \Pi_{\mathcal{Z}} [z_{k-1} - \eta F(w_k)], \qquad w_{k+1} = \Pi_{\mathcal{Z}} [z_k - \eta F(w_k)] \tag{1}$$

Clearly, the OG update rule is well defined for any operator $F$ and any closed convex set $\mathcal{Z}$, and this is how the OG algorithm is defined for variational inequalities.

**The Extragradient Algorithm.** Let $z_k$ be the action profile played at day $2k$ and assume that all players update their actions according to EG with step-size $\eta$. Then players play according to action profile $z_{k+\frac{1}{2}}$ at day $2k+1$ and action profile $z_{k+1}$ at day $2k+2$, where:

$$z_{k+\frac{1}{2}} = \Pi_{\mathcal{Z}} [z_k - \eta F(z_k)], \qquad z_{k+1} = \Pi_{\mathcal{Z}} \left[ z_k - \eta F(z_{k+\frac{1}{2}}) \right] \tag{2}$$

Similarly, the EG update rule is well defined for any operator $F$ and any closed convex set $\mathcal{Z}$. For the rest of the paper, we only consider the case where all players use the OG (or EG) algorithm with constant step-size $\eta$. We omit superscripts that denote players' identity and use Expression (1) (or Expression (2)) for the update rule when players use OG (or EG).

## 3 The Tangent Residual and Its Properties

We formally introduce our key performance measure, the tangent residual . We define the tangent residual over operators rather than games, as it will be easier to provide intuition behind its formulation.

**Definition 3** (Unit Normal Cone). *Given a closed convex set $\mathcal{Z} \subseteq \mathbb{R}^n$ and a point $z \in \mathcal{Z}$, we denote by $N_{\mathcal{Z}}(z) = \langle v \in \mathbb{R}^n : \langle v, z' - z \rangle \leq 0, \forall z' \in \mathcal{Z} \rangle$ the normal cone of $\mathcal{Z}$ at point $z$ and by $\widehat{N}_{\mathcal{Z}}(z) = \{v \in N_{\mathcal{Z}}(z) : \|v\| \leq 1\}$ the intersection of the unit ball with the the normal cone of $\mathcal{Z}$ at $z$. Note that $\widehat{N}_{\mathcal{Z}}(z)$ is nonempty and compact for any $z \in \mathcal{Z}$, as $(0, \ldots, 0) \in \widehat{N}_{\mathcal{Z}}(z)$.*

**Definition 4** (Tangent Residual). *Given an operator $F : \mathcal{Z} \to \mathbb{R}^n$ and a closed convex set $\mathcal{Z}$, let $T_{\mathcal{Z}}(z) := \{z' \in \mathbb{R}^n : \langle z', a \rangle \leq 0, \forall a \in N_{\mathcal{Z}}(z)\}$ be the tangent cone of $z$,[5] and define $J_{\mathcal{Z}}(z) := \{z\} + T_{\mathcal{Z}}(z)$. The tangent residual of $F$ at $z \in \mathcal{Z}$ is defined as $r_{(F,\mathcal{Z})}^{tan}(z) := \|\Pi_{J_{\mathcal{Z}}(z)}[z - F(z)] - z\|$. An equivalent definition is $r_{(F,\mathcal{Z})}^{tan}(z) := \sqrt{\|F(z)\|^2 - \max_{\substack{a \in \widehat{N}_{\mathcal{Z}}(z), \\ \langle F(z),a \rangle \leq 0}} \langle a, F(z) \rangle^2}$.*

**Remark 4.** *We show the equivalence of the two definitions of tangent residual in Lemma 5 in Appendix D. We may use either of them depending on which one is more convenient.*

When the convex set $\mathcal{Z}$ and the operator $F$ are clear from context, we are going to omit the subscript and denote the unit normal cone as $\widehat{N}(z) = \widehat{N}_{\mathcal{Z}}(z)$ and the tangent residual as $r^{tan}(z) = r_{(F,\mathcal{Z})}^{tan}(z)$. Although the definition is slightly technical, one can think of the tangent residual as the norm of another operator $\widehat{F}$, which is $F$ projected to all directions that are not "blocked" by the boundary of $\mathcal{Z}$ if one takes an infinitesimally small step $\epsilon \cdot F(z)$, which is the same as projecting $F$ to $J_{\mathcal{Z}}(z)$. Intuitively, if the tangent residual is small, then the next iterate will not be far away from the current one.

**Tangent Residual and Its Connection with Games.** Given a monotone game $\mathcal{G}$, we denote by $r_{\mathcal{G}}^{tan}(z) = r_{(F_{\mathcal{G}}, \mathcal{Z})}^{tan}(z)$. In the next lemma, we argue that a small tangent residual implies a

---

[5]Interested readers can find a thorough introduction of the tangent cone and its definition for general feasible sets in Chapter 6.A of [RW09]. When the feasible set is convex, Theorem 6.9 of [RW09] provides a more succinct definition of the tangent cone and normal cone. When the feasible set is a closed convex set, Corollary 6.30 of [RW09] further states that the tangent cone is the polar cone of the normal cone. As we consider closed convex sets in our paper, we choose to define the tangent cone for closed and convex sets directly as the polar cone of the normal cone, as it is the most convenient definition for us.

small gap and total gap, hence it suffices to show that the last-iterate has a small tangent residual. The proof is postponed to Appendix E.

**Lemma 2.** *[Adapted from Theorem 10 in [GPDO20] and Proposition 2 in [GPD20]] Let $\mathcal{G} = (\mathcal{N}, \{\mathcal{Z}^{(i)}\}_{i \in \mathcal{N}}, \{f^{(i)}\}_{i \in [N]})$ be a monotone game where $\{\mathcal{Z}^{(i)}\}_{i \in \mathcal{N}}$ are closed convex sets. For $z \in \mathcal{Z}$, we have $\mathrm{GAP}_{\mathcal{G},D}(z) \leq D \cdot r_{\mathcal{G}}^{tan}(z)$ and $\mathrm{TGAP}_{\mathcal{G},D}(z) \leq \mathrm{GAP}_{\mathcal{G},\sqrt{N}D}(z) \leq \sqrt{N}D \cdot r_{\mathcal{G}}^{tan}(z)$.*

## 4  Last-Iterate Convergence Rate for EG and OG

We prove the last-iterate convergence rate for EG and OG. We first describe our proof plan.

**Proof Plan.** Our analyses for both algorithms follow the same three-step procedure: (i) define a potential function that measures the proximity to a Nash equilibrium of the current iterate; (ii) prove a best-iterate convergence rate, that is, show that in $T$ steps, there exists one iterate whose potential function is small; (iii) show that the potential function is non-increasing, so the last-iterate is the best-iterate, and the best-iterate convergence rate becomes the last-iterate convergence rate.

The first major challenge we face is to choose the appropriate potential functions. In the unconstrained case, the central quantity is the norm of the operator $F_{\mathcal{G}}$. The key component of the analyses [LZMJ20, GPDO20, GPD20, GLG21] is to establish that the norm of the operator $F_{\mathcal{G}}$ at the last iterate (also the $T$-th iterate) is upper bounded by $O(\frac{1}{\sqrt{T}})$, which implies a $O(\frac{1}{\sqrt{T}})$ last-iterate convergence rate for the gap function. In the constrained setting, the norm of the operator is a poor choice to measure the proximity to a Nash equilibrium, as it can be far away from 0 even at a Nash equilibrium.

**Potential function for EG.** We use the **tangent residual** as the potential function for EG. Our starting point is to find a suitable generalization for the norm of the operator in the constrained setting. A standard generalization is the natural residual (Definition 7 in Appendix E), which takes the constraints into account and is guaranteed to converge to 0 at the Nash equilibrium. Unfortunately, we observe that the natural residual is *not monotonically decreasing* even in basic bilinear games, making it difficult to directly analyze. Similar non-monotonicity has been observed for several other natural performance measures such as $\|z_k - z_{k+1/2}\|$,[6] $\|z_k - z_{k+1}\|$, $\max_{z \in \mathcal{Z}} \langle F(z), z_k - z \rangle$ and $\max_{z \in \mathcal{Z}} \langle F(z_k), z_k - z \rangle$, leaving all these quantities unsuitable as a potential function. See more discussion in Appendix J. Indeed, tangent residual is the only natural generalization of the norm of the operator that is always monotone in our numerical experiments.

**Potential function for OG.** We choose the potential function $\Phi(z_k, w_k) = \|F(z_k) - F(w_k)\|^2 + r^{tan}(z_k)^2$ in our analysis for OG. The potential function can be interpreted as the squared tangent residual $r^{tan}(z_k)^2$ and an extra correction term $\|F(z_k) - F(w_k)\|^2$. The potential function is discovered via SOS programming, with more details in Section 4.2.

### 4.1  Best-Iterate Convergence

En route to establish the last-iterate convergence of EG and OG, we first show a weaker guarantee known as the best-iterate convergence. In Lemma 3 we show that when all the players use EG for $2 \cdot T$ steps (OG for $T$ steps), then there exists $t^* \in [T]$ where $r^{tan}(z_{t^*+1}) \leq O(\frac{1}{\sqrt{T}})$ ($\Phi(z_{t^*}, w_{t^*}) \leq O(\frac{1}{T})$ resp.). The proof for EG builds upon the proof for the best-iterate w.r.t. $\|z_k - z_{k+\frac{1}{2}}\|$ by [Kor76, FP07], while the proof for OG builds upon the best-iterate proof for $\|z_k - w_{k+1}\|$ by [WLZL21a, HIMM19]. The proof of Lemma 3 is in Appendix F.

**Lemma 3.** *Let $\mathcal{G} = (\mathcal{N}, \{\mathcal{Z}^{(i)}\}_{i \in \mathcal{N}}, \{f^{(i)}\}_{i \in [N]})$ be an L-smooth and monotone game where $\{\mathcal{Z}^{(i)}\}_{i \in \mathcal{N}}$ are closed convex sets and let $z^*$ be a Nash Equilibrium of $\mathcal{G}$. Assume that all the players*

---

[6] $\|z_k - z_{k+1/2}\|$ is proportional to the norm of the operator mapping introduced in [Dia20].

*update their actions using the EG algorithm with arbitrary starting action profile $z_0$ and step-size $\eta \in (0, \frac{1}{L})$. Then for and any $T > 0$, there exists $t^* \in [T]$ such that:*

$$r^{tan}(z_{t^*+1}) \leq \frac{1 + \eta L + (\eta L)^2}{\eta} \frac{1}{\sqrt{T}} \frac{\|z_0 - z^*\|}{\sqrt{1 - (\eta L)^2}}.$$

*Assume that all the players update their actions using the OG algorithm with arbitrary starting action profiles $z_0, w_0$ and step-size $\eta \in (0, \frac{1}{2L})$. Then for and any $T > 0$, there exists $t^* \in [T]$ such that:*

$$\Phi(z_{t^*}, w_{t^*}) \leq \frac{1}{\eta^2 \cdot T} \left( \frac{4 + 6\eta^4 L^4}{1 - 4\eta^2 L^2} \|z_0 - z^*\|^2 + \frac{16\eta^2 L^2 + 6\eta^4 L^4}{1 - 4\eta^2 L^2} \|w_0 - z_0\|^2 \right).$$

## 4.2 Monotonicity of the Potentials

In this section, we prove that $r^{tan}(z_k)$ ($\Phi(z_k, w_k)$ resp.) is non-increasing across iterates of EG (OG resp.), which, in combination with Lemma 3, implies the last-iterate convergence rate of smooth monotone games when all players update their actions using EG (OG resp.).

**SOS Programming.** Suppose we want to prove that a polynomial $g(x) \in \mathbb{R}[x]$ is non-negative over a semialgebraic set $\mathcal{S} = \{x : g_i(x) \leq 0, \forall i \in [M], h_i(x) = 0, \forall i \in [N]\}$, where each $g_i(x)$ ($h_i(x)$ resp.) is also a polynomial. One way is to construct a *certificate of non-negativity*, by providing a set of nonnegative coefficients $\{p_i \geq 0\}_{i \in [M]}$ and $\{q_i\}_{i \in [N]}$ such that $g(x) + \sum_{i \in [M]} p_i \cdot g_i(x) + \sum_{i \in [N]} q_i \cdot h_i(x)$ is a SOS polynomial. Surprisingly, if $g(x)$ is indeed non-negative over $\mathcal{S}$, a certificate of non-negativity always exists as guaranteed by a foundational result in real algebraic geometry – the Krivine-Stengle Positivestellensatz [Kri64, Ste74], a generalization of Artin's resolution of Hilbert's 17th problem [Art27]. The SOS programming [Nes00, Par00, Par03, Las01, Lau09] is a systematic way to search for such a certificate using semidefinite programming. See Appendix B for details.

Our approach is to apply SOS programming to search for a certificate of non-negativity for $r^{tan}(z_k)^2 - r^{tan}(z_{k+1})^2$ and $\Phi(z_k, w_k) - \Phi(z_{k+1}, w_{k+1})$ for every $k$, over a semialgebraic set defined by the update rule of the corresponding algorithm and set of constraints $\mathcal{Z}$.

Two important challenges with this approach is that the number of variables depends on the dimension of $\mathcal{Z}$ and there are infinitely many constraints associated with the problem (e.g. the set $\{\langle F(z) - F(z'), z - z'\rangle \geq 0, \forall z, z' \in \mathcal{Z}\}$). We provide a detailed exposition for the reduction of the number of constraints and the efficient formulation into a SOS program in Appendix G.3, where we prove the monotonicity of our potential function for EG in both the unconstrained and constrained settings.

**Searching for the potential functions.** The potential function of our analysis of OG is directly discovered using an SOS program. The program is formulated by searching over linear combinations of $\|F(z_k)\|^2 - \|F(z_{k+1})\|^2, \|F(w_k)\|^2 - \|F(w_{k+1})\|^2, \langle F(z_k), F(w_k)\rangle - \langle F(z_{k+1}), F(w_{k+1})\rangle$ and $r^{tan}(z_k)^2 - r^{tan}(z_{k+1})^2$, under (i) the constraint that the linear combination is non-increasing,[7] and (ii) the constraints induced by properties of the operator $F(\cdot)$, the update rule of OG and the set $\mathcal{Z}$ (See Appendix G.3 for a demonstration of the induced semialgebraic set of EG algorithm).

Our basis functions are chosen in a way so that we can search over all candidate potential functions that are a linear combination of (i) $r^{tan}(z_k)^2$ and (ii) any squared norms of a linear combination of $F(z_k), F(w_k)$. Observe that the difference between two consecutive iterates of any of the above functions can be expressed as a linear combination of the basis functions we chose.

We then use the linear combination output by the SOS program as the potential function in our analysis. We believe our heuristic for finding a potential function could be useful in

---

[7] To avoid finding the trivial linear combination, i.e., all coefficients equal to 0, we also use the objective function in the SOS program to encourage a non-trivial solution if one exists by, for example, maximizing the sum of the coefficients of the linear combination.

other settings. In general, one can first choose a collection of basis functions that may be part of a potential function, then use SOS programming to search over all linear combinations of the basis functions subject to the constraint that the linear combination is non-negative to discover the potential function.

Initially, our candidate potential functions included the squared tangent residuals evaluated at $z_k$ and $w_k$ and all degree-2 monomials of all vectors of interest. In other words, we searched over all linear combinations of (i) $r^{tan}(z_k)^2$, $r^{tan}(w_k)^2$ and (ii) any linear combination of degree-2 monomials of $z_k, F(z_k), w_k, F(w_k)$, subject to the constraint that the candidate potential function is non-negative. We included the tangent residuals in the basis of our potential functions due to their central role in the analysis of the EG algorithm (we show that the tangent residual is non-increasing across the iterates of the EG algorithm in Theorem 2).

We also observed that any non-zero linear combination of only the tangent residuals evaluated at $z_k$ and $w_k$ is not monotone for the OG algorithm. Motivated by this, we enlarged the basis to include the degree-2 monomials of $z_k, F(z_k), w_k, F(w_k)$ so that we had the flexibility to introduce an extra correction term in the potential function. Starting from this very flexible class of potentials functions, we gradually removed elements in the basis, and the basis that we selected in the paper was a minimal basis such that: (i) it contains a non-increasing potential function and (ii) it enjoys best-iterate convergence (See Lemma 17 in Appendix H.2) and (iii) the discovered potential function bounds the gap functions.

We establish the monotonicity of the potential functions in Theorem 1 and Theorem 2, which are both stated for general monotone operators over convex sets rather than games for technical convenience. We turn our attention back to games in Section 4.3 where we provide formal last-iterate convergence guarantees for smooth monotone games.

### 4.2.1 Monotonicity of $\Phi(z_k, w_k)$ for OG in the Unconstrained Setting

To better illustrate our approach, we prove in Theorem 1 the monotonicity of $\Phi(z_k, w_k)$ for the OG algorithm in the unconstrained setting. Note that this is already a strengthening of [GPD20], where we provide a simple potential function that can be used to directly argue that $\|F(z_k)\|^2 = O(\frac{1}{k})$ without making any second-order smoothness assumption.

**Theorem 1.** *Let $F : \mathcal{Z} \to \mathbb{R}^n$ be a monotone and $L$-Lipschitz operator. For any $z_k, w_k \in \mathbb{R}^n$, the unconstrained OG algorithm with step-size $\eta \in (0, \frac{1}{2L})$ satisfies $\Phi(z_k, w_k) \geq \Phi(z_{k+1}, w_{k+1})$.* [8]

*Proof.* Since $F$ is monotone and $L$-Lipschitz, we have $\langle F(z_{k+1}) - F(z_k), z_k - z_{k+1} \rangle \leq 0$ and $\|F(w_{k+1}) - F(z_{k+1})\|^2 - L^2\|w_{k+1} - z_{k+1}\|^2 \leq 0$. We simplify them using the update rule of OG and $\eta^2 L^2 < \frac{1}{4}$. In particular, we replace $z_k - z_{k+1}$ by $\eta F(w_{k+1})$ and $w_{k+1} - z_{k+1}$ with $\eta F(w_{k+1}) - \eta F(w_k)$.

$$\langle F(z_{k+1}) - F(z_k), F(w_{k+1}) \rangle \leq 0, \tag{3}$$

$$\|F(w_{k+1}) - F(z_{k+1})\|^2 - \frac{1}{4}\|F(w_{k+1}) - F(w_k)\|^2 \leq 0. \tag{4}$$

It is not hard to verify the following identity:

$$\|F(z_k) - F(w_k)\|^2 + \|F(z_k)\|^2 - \|F(z_{k+1}) - F(w_{k+1})\|^2 - \|F(z_{k+1})\|^2$$

$$+ 2 \cdot \text{LHS of Inequality}(3) + 2 \cdot \text{LHS of Inequality}(4) = \frac{1}{2}\|F(w_k) + F(w_{k+1}) - 2F(z_k))\|^2.$$

Thus, $\|F(z_k) - F(w_k)\|^2 + \|F(z_k)\|^2 \geq \|F(z_{k+1}) - F(w_{k+1})\|^2 + \|F(z_{k+1})\|^2$. $\qquad\square$

### 4.2.2 Constrained EG and OG

In the constrained setting, the approach is more complicated due to the projections and we postpone any exposition to Appendix G.3. We state the results in the following theorem.

---

[8]In the unconstrained setting, $r^{tan}_{(F, \mathbb{R}^n)}(z) = \|F(z)\|^2$.

**Theorem 2.** *Let $\mathcal{Z} \subseteq \mathbb{R}^n$ be a closed convex set and $F : \mathcal{Z} \to \mathbb{R}^n$ be a monotone and L-Lipschitz operator. For any $z_k \in \mathcal{Z}$, the EG algorithm with step-size $\eta \in (0, \frac{1}{L})$ satisfies $r^{tan}(z_k) \geq r^{tan}(z_{k+1})$. Moereover, for any $z_k, w_k \in \mathcal{Z}$, the OG algorithm with step-size $\eta \in (0, \frac{1}{2L})$ satisfies $\Phi(z_k, w_k) \geq \Phi(z_{k+1}, w_{k+1})$, where $\Phi(z_k, w_k) = \|F(z_k) - F(w_k)\|^2 + r^{tan}(z_k)^2$ for all k.*

### 4.3 Last-Iterate Convergence of EG and OG

In this section, we formally combine Lemma 3 and Theorem 2 to show the last-iterate convergence of EG and OG with respect to the tangent residual, gap function, and the total gap function. Recall that when all the players update their actions using the EG algorithm, then $z_k$ is the action profile played at day $2 \cdot k$ and $z_{k+\frac{1}{2}}$ is the action played at day $2 \cdot k + 1$, while when all the players update their action profile using the OG algorithm, then $w_k$ is the action profile they play at day $k$. The formal proof of Theorem 3 is postponed at Appendix F.

**Theorem 3.** *Let $\mathcal{G} = (\mathcal{N}, \{\mathcal{Z}^{(i)}\}_{i \in \mathcal{N}}, \{f^{(i)}\}_{i \in [N]})$ be an L-smooth and monotone game where $\{\mathcal{Z}^{(i)}\}_{i \in \mathcal{N}}$ are closed convex sets and let $z^*$ be a Nash Equilibrium of $\mathcal{G}$. Assume that all the players update their actions using the EG algorithm with arbitrary starting action profile $z_0$ and step-size $\eta \in (0, \frac{1}{L})$. Let $D_0 = \frac{3\|z_0 - z^*\|}{\sqrt{1 - (\eta L)^2}}$, then for and any $T > 0$ and $D > 0$,*

$$\max\left(r^{tan}(z_T), \frac{\text{GAP}(z_T)}{D}, \frac{\text{TGAP}(z_T)}{\sqrt{N} \cdot D}\right) \leq \frac{D_0}{\eta\sqrt{T}},$$

$$\max\left(r^{tan}(z_{T+\frac{1}{2}}), \frac{\text{GAP}(z_{T+\frac{1}{2}})}{D}, \frac{\text{TGAP}(z_{T+\frac{1}{2}})}{\sqrt{N} \cdot D}\right) \leq \frac{(1 + \eta L) \cdot D_0}{\eta\sqrt{T}}.$$

*When all the players update their actions using the OG algorithm with arbitrary starting action profiles $z_0, w_0$ and step-size $\eta \in (0, \frac{1}{2L})$. Denote $D_0 := \frac{\sqrt{2}(2 + \eta L)}{\sqrt{1 - 4 \cdot (\eta L)^2}} \cdot \sqrt{(4 + 6\eta^4 L^4)\|z_0 - z^*\|^2 + (16\eta^2 L^2 + 6\eta^4 L^4)\|w_0 - z_0\|^2}$. Then for any $T > 0$ and $D > 0$,*

$$\max\left(r^{tan}(w_{T+1}), \frac{\text{GAP}(w_{T+1})}{D}, \frac{\text{TGAP}(w_{T+1})}{\sqrt{N} \cdot D}\right) \leq \frac{D_0}{\eta\sqrt{T}}.$$

For EG, according to our bound in Theorem 3, the optimal value of $\eta$ is $\frac{1}{\sqrt{2}L}$, which implies that for any $D \geq 0$, $\text{GAP}(z_T) \leq \frac{6\|z_0 - z^*\|DL}{\sqrt{T}}$, and when $D \geq \|z_0 - z^*\|$, then $\text{GAP}(z_T) \leq \frac{6D^2L}{\sqrt{T}}$; For OG initialized with $w_0 = z_0$, by numerically optimizing the bound on Theorem 3, we set $\eta$ to be $\frac{0.34}{L}$, in which case for $D \geq \|z_0 - z^*\|$, $\text{GAP}(w_{T+1}) \leq \frac{26.82D^2L}{\sqrt{T}}$. Both upper bounds matches the $\Omega(\frac{D^2L}{\sqrt{T}})$ lower bound for EG, OG, and more generally all p-SCLI algorithms [GPDO20, GPD20] in terms of the dependence on $D$, $L$, and $T$.

In Appendix I we show that EG and OG algorithm also have last-iterate convergence for monotone variational inequalities with respect to several standard performance measures (See Appendix D for the formal definition of monotone variational inequality).

## 5 Conclusion

We provide the first and tight last-iterate convergence rate of the EG and OG algorithm for smooth monotone games and Lipschitzs and monotone VIs. Our proof is based on the tangent residual, which is a new proximity measure to a Nash equilibrium, and an accompanying SOS-based analysis. We believe our techniques may be useful in the study of last-iterate convergence for other algorithms.

**Acknowledgement:** This work was supported by a Sloan Foundation Research Fellowship and the NSF Award CCF-1942583 (CAREER). Part of this work was done while the authors were visiting the Simons Institute for the Theory of Computing.

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
