# OpenReview forum: "Finite-Time Last-Iterate Convergence for Learning in Multi-Player Games"
_NeurIPS.cc/2022/Conference — NeurIPS 2022 Accept_

### Official Review · Reviewer_bUVX · 2022-07-11

**Rating:** 8
**Confidence:** 2
**Soundness:** 3 good
**Presentation:** 4 excellent
**Contribution:** 4 excellent

**Summary:**

This paper provides the first last-iterate convergence rates of extragradient algorithm (EG) or the optimistic gradient algorithm (OG) to Nash equilibrium of constrained smooth monotone games in terms of the gap function and Lipschitzs and monotone variational inequality, which match the existing lower bounds $\mathcal{O}(T^{-1/2})$. Tangent residual is proposed as a new proximity measure to a Nash equilibrium and used for convergence analysis.

**Questions:**

(1) Why does contribution 1 on page 3 not include EG?

(2) Typo in Table 1: "cocoercive".

(3) Typo in Line 95: "require**s** the setting to be unconstrained".

(4) In Definition 1, you might use parentheses like (that is, ...), (i.e. ...), since at first sight, I misunderstood the formula after i.e. as equivalent to the condition that $F_{\mathcal{G}}$ is both monotone and Lipschitz. Also,  $F_{\mathcal{G}}$ is monotone if and only if $f^{(i)}$ is convex for any $i$, yes? If yes, You might add that intuition.

(5) In Definition 2, both gap functions are non-negative, and become 0 if and only if $z$ is a Nash equilibrium, right? If yes, you might write down this intuition to help readers outside this area understand. Also, a larger (total) gap function value indicates a larger distance (and thus smaller proximity) to Nash equilibrium, right? If yes, I think it better to describe the gap functions as distance measures (instead of proximity) to Nash equilibrium.

(6) Is it convenient to clarify the intuition of the notions in Definitions 3?

(7) $r_{\mathcal{G}}^{\text{nat}}$ appears in line 230 for the first time so here you may cite Definition 7 in Appendix E. (In Line 254, it should be Appendix E, not D.)

(8) Are the potential differences $r^{tan}(z_k)^2-r^{tan}(z_{k+1})^2$ and $\Phi(z_k,w_k)-\Phi(z_{k+1},w_{k+1})$ polynomial? If not, how can you apply SOS programming to them?

(9) It seems better to compute for more iterations in the experiments and plot the potential function value and the performance measures in Appendix J, to show that all of them approach 0 (yes?), and only the potential function value decreases.

**Ethics Review Area:**

["I don’t know"]

**Limitations:**

In the checklist, item 1b said "Did you describe the limitations of your work? [Yes] Our results are theoretical. We state the assumptions of our results in the statements of the lemmas." Do you consider the assumptions as limitations? If yes, do you intend to weaken them in future works?

The end of the introduction also proposes a future direction to apply the techniques of this paper to stochastic gradient feedback, which implies a limitation and its solution.

**Strengths And Weaknesses:**

Originality: The algorithms and problem settings exist in the literature, but the non-asymptotic convergence rate in constrained smooth monotone games is new and tight, and the notion of tangent residual and the choice of potential functions for convergence analysis are novel. Hence, I think the originality is sufficient.

Quality: This is a complete piece of work with both strengths and weaknesses. The claims are well supported by the theoretical analysis which looks reasonable, elaborate, and sophisticated (I have read the Lemmas and theorems but have not checked the proof). The experiments look reasonable to me, but it seems that the numerical evidence can be stronger if the authors compute for more iterations in the experiments and plot the potential function value and the performance measures in Appendix J. (see my final question for detail).

Clarity: The submission is very clear and well organized. The intuition behind some notions could be clarified as elaborated in my questions.

Significance: The results look important to me since it provides the first non-asymptotic convergence rate in constrained smooth monotone games, which also matches the lower bound.

---

> ### Author Response · Authors · 2022-08-02
> **Thank you for your Review (Part 2 of 2)**
>
> **Question (6):** We first provide some intuition behind the definition of the normal cone to a closed convex set $\mathcal{Z}$ at point $z\in\mathcal{Z}$, i.e., $N_{\mathcal{Z}}(z) := \{ u: \langle u, z' - z \rangle \le 0, \forall z' \in \mathcal{Z}\}$. Geometrically, this means that for any vector $w\in N_{\mathcal{Z}}(z)$ and any point $\overline{z}\in \mathcal{Z}$, the angle between vector $w$ and $\overline{z}-z$ is always obtuse. Additionally, this also implies that the projection of $z+w$ to set $\mathcal{Z}$ is exactly $z$.
>
>  As the normal cone $N_{\mathcal{Z}}(z)$ could be unbounded, we further define the unit normal cone, which is the intersection of the normal cone with the unit ball. The unit normal cone is  closed and bounded for any $z\in \mathcal{Z}$, which is convenient to work with in some of our analysis.
>
>
> **Question (8):** Although not obvious, both of the potential differences can indeed be represented as degree-2 polynomials.
> Lemma 13 in Appendix G.3.2 (line 969-973) shows how to represent the squared tangent residual $r^{tan}(z_k)^2$ as a degree-2 polynomial over the variable $F(z)$ and an auxiliary variable $c(z)$. The SOS program that we use to certify the non-negativity of the potential difference $r^{tan}(z_k)^2-r^{tan}(z_{k+1})^2$ can be found in line 1003-1006. We explain why this SOS program captures the potential difference in line 984-1002. In a similar manner, we can represent the potential difference $\Phi(z_k,w_k) - \Phi(z_{k+1},w_{k+1})$ as a degree-2 polynomial over a semi-algebraic set and capture it with a SOS program.
>
> **Question (9):** Thank you for your suggestion. As you correctly observed, all of these quantities eventually converge to $0$ but some are not monotonically decreasing. The only monotone quantity is our potential functions.
>
> We have uploaded a revised supplementary material that includes numerical experiments with more iterations in Appendix J. More specifically, in Figures 6-10 we plot the values of the non-monotone performance measures of interest as well as the tangent residual (properly scaled so that it can fit in the figure) for more iterations using the same instances as provided Appendix J with starting point $z_0 = (0.25,0.25,0.25,0.25)^{T}$ and step size $\eta = 0.1$. We will include these plots in the updated version.
>
>
> **About Limitations:**
>
> Our result provides the tight last-iterate convergence rate in monotone games and monotone variational inequalities with deterministic gradient feedback. There are two assumptions/limitations on the setting we considered, and we plan to weaken them. The first assumption is that the game is monotone, while there are many interesting and relevant non-monotone games. A concrete example is the class of games that satisfy the weak MVI condition as mentioned by reviewer x4Lr. Another example is two-player zero-sum Markov games. Additionally, we  hope to extend the result to handle stochastic gradient feedback (also mentioned by reviewer ZxuX), thus eliminating the deterministic gradient feedback assumption. We consider removing either of the two assumptions important and interesting. We are currently  working on both of them. We will include more discussion about these assumptions/limitations in the updated version of the paper.

---

> > ### Comment · Reviewer_bUVX · 2022-08-08
> > **Reviewer bUVX is satisified and will keep rating 8.**
> >
> > Reviewer bUVX is generally satisified with the authors' rebuttal and will keep rating 8.
> >
> > For Q1, your contribution 2 includes both EG and OG. Are they both no-regret for solving Lipschitz and monotone VIs?
> > For Q9, in the plots in Appendix J, you could indicate the meaning of different colors using legends.
> >
> > Thank you.

---

> > > ### Author Response · Authors · 2022-08-09
> > > **Thank you for the response**
> > >
> > > Thank you for the encouraging comment.
> > >
> > > Regarding Q1, for monotone and Lipschitz VI's our focus is algorithmic, and the goal is to understand the convergence rates of these algorithms rather than viewing them as game dynamics. Thus in contribution 2, we include both EG and OG regardless of whether they have the extra no-regret property. We will add more discussion to clarify in the next version.
> > >
> > > Regarding Q9, we will add legends in figures. Thank you for the suggestion.

---

> > > > ### Comment · Reviewer_bUVX · 2022-08-09
> > > > **OK. Thanks.**
> > > >
> > > > OK. Thanks.

---

> ### Author Response · Authors · 2022-08-02
> **Thank you for your review (Part 1 of 2)**
>
> Thank you for your thoughtful and encouraging review.
>
> **Question (2), (3), and (7):** Thank you for pointing them out. We will fix them in the next version of the paper.
>
>
> **Question (1):**
> We did not include the EG algorithm in contribution 1 because EG is not a no-regret algorithm (See Proposition 10 in [1]). We analyze EG, as it is one of the most classical and popular algorithms for solving monotone variational inequalities, and its last-iterate convergence rate in the constrained setting was an open problem recognized in [1-4].
>
>
> **Question (4):** Thank you for the suggestion. We will rephrase Definition 1 to make it more clear. We will also add a discussion to clarify the relationship between monotone games and concave games (i.e., each player's cost function $f^{(i)}$ is convex for any $i$). When $F_\mathcal{G}$ is monotone, each $f^{i}(z_i, z_{-i})$ is convex in $z_i$ for every fixed $z_{-i}$. The converse statement does not hold in general. Consider a two player $2$-dimension game $\mathcal{G}$ where the cost functions $f^{1}(z_1,z_2) = f^{2}(z_1,z_2) = z_1 z_2$. Clearly, $f^{1}(z_1,z_2)$ (or $f^{2}(z_1,z_2)$) is convex in $z_1$ (or $z_2$) if we fix $z_2$ (or $z_1$). It is not hard to see that  $F_\mathcal{G}(x,y) = (y, x)$ for any $(x,y) \in \mathbb{R}^2$. Therefore, the game is not monotone as $\langle F_\mathcal{G}(z_1, z_2)- F_\mathcal{G}(z_2, z_1), (z_1,z_2) - (z_2,z_1)\rangle = -2(z_1 -z_2)^2 < 0$ for any $z_1 \ne z_2$.
>
>
>
>
> **Question (5):** Thank you for the suggestion. You are right that both gap functions are always non-negative and equal to $0$ only at a Nash equilibrium. The total gap function
> is the sum of the suboptimality gaps, i.e., a player $i$'s suboptimality gap is her cost under $z$ minus her cost after best responding to $z_{-i}$ within $\mathcal{B}(z_i,D)$. As the suboptimality gap is nonnegative for every player, a small total gap implies that no player can deviate from their current action to significantly improve their cost, and the action profile is close to a Nash equilibrium. The gap function with radius $D$ is an upper bound of the total gap with radius $\frac{D}{\sqrt{N}}$, where $N$ is the number of players, as shown in line 788-789, so if an action profile has small gap function, then again no player can deviate to significantly reduce their cost. We will include the relationship between the gap function and the total gap function in the statement of Lemma 2. We will also include more discussion regarding the intuition behind the definitions in the next version of the paper.
>
> Finally, we would like to clarify the relationship between the gap function and the distance to a Nash equilibrium. Here we provide an example where a smaller total gap (or gap) does not imply that it is closer to an equilibrium in terms of the *Euclidean distance*, although the smaller total gap (or gap) does mean that the gain from deviation is smaller for every player, and thus it is closer to a Nash equilibrium in this sense.
>
> Example: Consider the monotone two-player game, where $\mathcal{Z}_1=\mathcal{Z}_2=[-10,10]^2$ and $f_1(z_1,z_2)=-f_2(z_1,z_2)=z_1^T A z_2$, where $A$ is a $2\times 2$ matrix whose first row is $\begin{array} {cc} (1 & 0)\end{array}$ and second row is  $\begin{array} {cc} (0 & 20)\end{array}$. It is not hard to verify that the unique Nash Equilibrium is $z^* =[ (0,0),(0,0)]$. Consider action profiles $z= [(10,0),(0,0)]$ and $\overline{z}=[(0,1), (0,0)]$.
> Observe that $| z-z^* |=10> |\overline{z}- z^* |=1$,
> while $Gap(z) = 100< Gap(\overline{z})=200$. Note that in this setting, the gap function is equivalent to the total gap function, so this example also shows that a smaller total gap does not imply a smaller distance to the equilibrium.

---

### Official Review · Reviewer_x4Lr · 2022-07-13

**Rating:** 7
**Confidence:** 4
**Soundness:** 4 excellent
**Presentation:** 4 excellent
**Contribution:** 4 excellent

**Summary:**

The paper focuses on proving last iterate convergence for both Extra Gradient and Optimistic Gradient Descent methods for monotone games in both constrained and unconstrained settings (these setting captures convex-concave min-max optimization for example where the operator is monotone). The authors show tight last-iterate convergence of $O(1/\sqrt{T})$ without any uniqueness assumptions that prior works had consider (without proving rates). Please note that there were works that established last iterate convergence that the dependence on the parameters of the game were not clear (it might be exponential for example in the size of the game). The authors show dependence that is polynomial in the parameters of the game (Lipschitz constant and diameter of the domain for constrained). The idea is first to establish best iterate convergence and then show that the tangent residual (effectively the Hamiltonian) is a potential function.

**Questions:**

My only question is if these results can be generalized for weak MVI type functions/problems.

**Limitations:**

The work is theoretical.

**Strengths And Weaknesses:**

Pros:
1) The paper is well written and the proof is elegant.
2) The problem was an open-question that researchers were actively working on including the reviewer.
3) The result does not hide any exponential dependence on the size of the game, it is very clean with normal assumptions.

Cons:
1) The idea of using sum of squares was already in the literature to construct/prove potential functions. This fact though does not decrease the significance of this work.

---

> ### Author Response · Authors · 2022-08-02
> **Thank you for your Review**
>
> Thank you for your thoughtful and encouraging review. We totally agree with you that obtaining
> last-iterate convergence rate for weak MVI type problems is a very important and interesting question. Our results do not readily extend to the weak MVI setting. Indeed, since there exist best-iterate convergence rates for weak MVI type problems, we hope that we could again find a potential function that allows us to strengthen the best-iterate convergence rate to last-iterate convergence rate, as we did in this paper. However, we have yet to find such a potential function. We still feel optimistic that our SOS-based approach will be useful here, perhaps in combination with other tools/techniques. Overall, we believe our SOS-based approach is powerful and could find applications in other problems in multi-agent learning and the design and analysis of iterative methods.

---

> > ### Comment · Reviewer_x4Lr · 2022-08-07
> > **Thank you for the feedback**
> >
> > I feel the difference between last iterate and best iterate has to do with uniqueness and non-uniqueness of the underlying variational inequality problem. I feel the paper should get a spotlight, I will continue the discussion with fellow reviewers and AC.

---

> > > ### Author Response · Authors · 2022-08-09
> > > **Thank you for the response**
> > >
> > > Thank you for the encouraging comment and the helpful insight. Please don't hesitate to contact us should you have any further questions.

---

### Official Review · Reviewer_ZxuX · 2022-07-15

**Rating:** 7
**Confidence:** 4
**Soundness:** 3 good
**Presentation:** 3 good
**Contribution:** 3 good

**Summary:**

This paper studies the last-iterate convergence rate to a Nash equilibrium in monotone games (with gradient feedback). It extends previous work to analyze the convergence of no-regret learning algorithm with constant step size in (possibly) constrained games. The authors propose a novel notion, namely the tangent residual, and use it to discover the potential function to show that the extragradient algorithm and the optimistic gradient algorithm have tight last-iterate convergence rates.

**Questions:**

- On the search for the potential functions: are the potentials discovered by the SOS program unique (given a set of basis functions)? If not, do we have a solution selection problem (i.e., different potential might provide different rates?)
Can you please give some intuition of the choice of the basis functions in line 302 (or this choice simply comes from the needs in the proof)?
What is the running time/ complexity of SOS programing (e.g., w.r.t. the number of basis functions)?

- On parameters tuning: Is the parameter $eta$ in Lines 349 and 350 are chosen to give the optimal constants in the convergence rates? If not, can we do it (and get rid of the bigO notation)? Technically, the choice of D can be any upper-bound of the one given in Lines 349 and 350, right? Notably, we can also use the popular choice as stated in footnote 4?

- Perspective: can this line of work be extended to stochastic gradient feedback case? (see e.g., discussion in Appendix A.4, of [GPD20])


**Limitations:**

the authors adequately addressed the limitations and potential societal impact of their work

**Strengths And Weaknesses:**

Strengths: The paper is well-written. The achieved results are quite interesting and relevant. The proofs and appendix are very detailed and provide strong supports for the claimed results. The tangent residual notion and the proposed computer-aided proof technique are, as far as I know, novel in this research context; moreover, they might be useful beyond the scope of this paper.

Weaknesses: no major weaknesses. Optimally tuning parameters of the algorithms (and explicitly determining the constants in convergence rates) might make the results clearer.

---

> ### Author Response · Authors · 2022-08-02
> **Thank you for your Review - Part 1 of 2**
>
> Thank you for your thoughtful and encouraging review. Please see our responses to your questions as follows:
>
>
> **On the search for the potential functions:** Our potential function $\Phi(z_k,w_k)$ is sufficient to obtain a convergence rate that is tight up to a constant factor. However, we are unsure whether this potential function is the unique non-increasing and non-trivial function in the constrained setting.
>
> In our case, to avoid the trivial all zero solution, we set up our SOS program to optimize an objective over a random linear combination of the coefficients of the basis functions. We ran our SOS program many times with a newly generated random objective each time, and the discovered potential function was the same in every execution (modulo small numerical noises). However, it might be possible to choose a different set of basis functions to obtain a different potential function.
> It is also conceivable that a different potential function could allow us to improve the convergence rate, but not more than a constant factor.
> More specifically, our convergence rates are already tight with respect to $D$, the Lipschitz constant $L$, and the number of iterations $T$ (See Lines 349-352), so a different potential function could at best be used to sharpen the constant in the convergence rate.
>
>
>
> Our basis functions are chosen in a way so that we can search over all candidate potential functions that are a linear combination of (i) $r^{tan}(z_k)^2$ and (ii) any squared norms of a linear combination of $\{F(z_k), F(w_k)\}$.
> Observe that the difference between two consecutive iterates of any of the above functions can be expressed as a linear combination of the basis functions we chose.
>
>
> *Intuition behind the choice of our basis*:
> Initially, our candidate potential functions included the tangent residuals evaluated at $z_k$ and $w_k$ and all degree-2 monomials of all vectors of interest. In other words, we searched over all linear combinations of (i) $\{r^{tan}(z_k)^2, r^{tan}(w_k)^2\}$ and (ii) any  linear combination of degree-2 monomials of $\{ z_k, F(z_k), w_k, F(w_k) \}$, subject to the constraint that the candidate potential function is non-negative.
> We included the tangent residuals in the basis of our potential functions due to their central role in the analysis of the EG algorithm (we show that the tangent residual is non-increasing across the iterates of the EG algorithm in Theorem 2).
>
> We also observed that any non-zero linear combination of only the tangent residuals evaluated at $z_k$ and $w_k$ is not monotone for the OG algorithm. Motivated by this, we enlarged the basis to include the degree-2 monomials of $\{z_k, F(z_k), w_k, F(w_k)\}$ so that we had the flexibility to introduce an extra correction term in the potential function.
> Starting from this very flexible class of potentials functions,
> we gradually removed elements in the basis, and the basis that we selected in the paper was a minimal basis such that: (i) it contains a non-increasing potential function and (ii) it enjoys best-iterate convergence (See  Lemma 17 in Appendix H.2) and (iii) the discovered potential function bounds the gap functions. We will make sure to include a discussion about the intuition in the next version of the paper.

---

> > ### Author Response · Authors · 2022-08-02
> > **Thank you for your Review - Part 2 of 2**
> >
> > *Running time/complexity of the SOS programs:* As we are only solving degree-$2$ SOS programs, we can turn it into a semidefinite program (SDP) with (i) $O(C)$  constraints, where $C$ is the number of constraints in the semialgebraic set (see line 304-306 and Appendix G.3 for the semialgebraic set for EG); (ii) $O(V^2+B+C)$ variables, where $V$ is the total number of variables that appeared in the basis functions, and $B$ is the number of basis functions.
> > In our case, $B=4$, $C=27$ and $V=10$. We solved the SOS programs in MATLAB using the SOSTOOLS package. The SOS program that searches and certifies the potential function in Theorem 7 took less than 1 minute to run using MATLAB online. The degree-2 SOS programs we used to certify the monotonicity of the proposed potential functions in Theorem 4 and Theorem 5 also took less than 1 minute to run using MATLAB online.
> >
> >
> >
> >
> >
> > **On parameters tuning:**
> > We would like to note that there is a typo in lines 349 and 350.
> > In the revised supplementary material, we corrected it as following:
> >
> >
> > ''For EG, choosing $\eta$ to be $\frac{1}{2L}$ and $D\geq |z_0-z^* |$,
> > then $GAP(z_T)=O(\frac{D^2L}{T})$ ;
> > For OG, choosing $\eta$ to be $\frac{1}{3L}$ and $D \geq max ( |z_0-z^* |,  |z_0-w_0 | )$, then $GAP(w_{T+1})=O\left(\frac{D^2L}{\sqrt{T}}\right)$.''
> >
> >
> >
> > The step-size $\eta$'s in the above sentence are not chosen to optimize the constants in the convergence rates.  Theorem 3 holds for any choice $D > 0$ or any of the popular choices as mentioned in footnote 4.
> > For example, when we run the EG algorithm with step-size $\eta$ on an $L$ Lipschitz operator for $T$ iterations, then when $D>0$, $GAP(z_T) \leq \frac{3 |z_0-z^*|}{\eta L \sqrt{1-(\eta L)^2}} \frac{L D}{\sqrt{T}}$ and when $D\geq |z_0-z^*|$,
> > $GAP(z_T)\leq \frac{3}{\eta L \sqrt{1-(\eta L)^2}}\frac{L D^2}{\sqrt{T}}$ (Similar to the bound in the lines 349 and 350).
> > By setting $\eta=\frac{1}{\sqrt{2}L}$,
> > we can get the optimal constant in the convergence rate of the EG based on our analysis, in which case for any $D>0$, $GAP(z_T)\leq 6|z_0-z^*|\frac{L D}{\sqrt{T}}$ and when $D\geq |z_0-z^*|$,
> > $GAP(z_T)\leq 6\frac{L D^2}{\sqrt{T}}$.
> > Similarly, for the OG algorithm initialized with $w_0=z_0$, the upper bound on the duality gap for $D\geq  |z_0-z^* |$ is $Gap(w_{T+1}) \le \frac{\sqrt{2}(2+\eta L)\sqrt{4+6(\eta L)^4}}{\eta L\sqrt{1-4(\eta L)^2}}\frac{ |z_0-z^* |DL}{\sqrt{T}}\leq \frac{\sqrt{2}(2+\eta L)\sqrt{4+6(\eta L)^4}}{\eta L\sqrt{1-4(\eta L)^2}}\frac{D^2L}{\sqrt{T}}$.
> > By numerically minimizing the factor $\frac{\sqrt{2}(2+\eta L)\sqrt{4+6(\eta L)^4}}{\eta L\sqrt{1-4(\eta L)^2}}$ for $\eta \in \left(0,\frac{1}{2L}\right)$,
> > we can calculate that the approximately optimal step-size according to our analysis is $\eta = \frac{0.34}{L}$,
> > with convergence rate $Gap(w_{T+1}) \le 26.82\cdot \frac{D^2L}{\sqrt{T}}$.
> > We can similarly get the optimal convergence rates without the big $\mathcal{O}$ for the other performance measure of interest by optimizing over the choice of step-size. We will include these optimal choices of step-size in the updated version of our paper.
> >
> >
> >
> >
> > **Perspective:** Extending the results to the stochastic setting is  a very interesting direction, and is indeed one of our next steps. We consider deriving last-iterate convergence rates under both the absolute random noise model and the relative random noise model important open problems. At this moment, we do not see how to directly extend our results to stochastic gradient feedback case. We are actively working on the stochastic case, and we are hopeful that our approach will be useful.

---

> > > ### Comment · Reviewer_ZxuX · 2022-08-08
> > > **Response to authors**
> > >
> > > Thank you for answering the questions and clarifying the details. I am satisfied with the answers and would like to keep the previously assigned score.
> > >
> > > A quick note: since you use it, it might be fair to cite the SOSTOOLS package:
> > > author = {A. Papachristodoulou, J. Anderson, G. Valmorbida, S. Prajna, P. Seiler, P. A. Parrilo, M. M. Peet and D. Jagt},
> > > title = {{SOSTOOLS}: Sum of squares optimization toolbox for {MATLAB}}
> > >
> > > Thanks again.

---

> > > > ### Author Response · Authors · 2022-08-09
> > > > **Thank you for the response**
> > > >
> > > > Thank you for the recommendation. We will include a reference for the SOSTOOLS package.

---

### Official Review · Reviewer_3NQG · 2022-07-18

**Rating:** 7
**Confidence:** 3
**Soundness:** 4 excellent
**Presentation:** 4 excellent
**Contribution:** 3 good

**Summary:**

This paper studies the last iterate rate of convergence of extra gradient and optimistic gradient algorithms in smooth monotone games with continuous convex action sets. More specifically, the authors prove that if players employ those algorithms with a constant step size they achieve last iterate convergence rate $O(1/\sqrt{T})$.

**Questions:**

I would like to ask about the definition provided in this paper for the tangent cone. I am not familiar with this definition, could the authors provide some reference for it?


**Limitations:**

This paper has no negative societal impact.

**Strengths And Weaknesses:**

This paper is well written and clear. The technique used for finding the potential function needed to prove the result is novel and interesting.
I consider the contribution of this paper clear and quite important, since it complements and extends existing results from the unconstrained to the constrained setting.


Minor comments: In line 231 there is a typo - hence (an) it suffices.
Line 276: action -> action.

---

> ### Author Response · Authors · 2022-08-02
> **Thanks for your Review**
>
> Thank you for your thoughtful and encouraging review.
> Regarding the question on the definition of the tangent cone, Chapter 6.A of [1] provides a thorough introduction of the tangent cone and its definition for general feasible sets. When the feasible set is convex, Theorem 6.9 of [1] provides a more succinct definition of the tangent cone and normal cone. When the feasible set is a *closed* convex set, Corollary 6.30 of [1] further states that the tangent cone is the polar cone of the normal cone. As we consider closed convex sets in our paper, we choose to define the tangent cone (Definition 4 of our paper) for closed and convex sets directly as the polar cone of the normal cone, as it is the most convenient definition for us. We will make sure to include additional discussion and the aforementioned reference about the tangent cone in the updated version of the paper.
>
>
> References:
>
> [1] Rockafellar, R. Tyrrell, and Roger J-B. Wets. Variational analysis. Vol. 317. Springer Science \& Business Media, 2009.

---

> > ### Comment · Reviewer_3NQG · 2022-08-07
> > **Response to authors**
> >
> > Thank you! I will take a look.

---

> > > ### Author Response · Authors · 2022-08-09
> > > **Thank you for the response**
> > >
> > > Dear reviewer,
> > > You are welcome. Please don't hesitate to contact us should you have any further questions.

---

### Author Response · Authors · 2022-08-02
**Response to all the reviewers**

We want to thank all the reviewers for constructive feedback, which helps us improve our paper. Please find below the response to each comment.

---

### Meta-Review · Area_Chair_LRYw · 2022-08-22

**Recommendation:** Accept
**Confidence:** Certain

**Metareview:**

This paper studies the last iterate rate of convergence of the well-known extragradient and optimistic gradient algorithms in smooth monotone games with continuous convex action sets. The main result of the paper is to show that both algorithms (with constant step size) enjoy tight last-iterate convergence rates for setting (previous papers either 1) only applied to unconstrained domains 2) were asymptotic, or 3) required dependence on arbitrarily large problem-dependent constants).

This paper resolves a well-known open problem within the min-max optimization community, and is likely to have significant impact. The reviewers agree that the paper is well-written, and the the techniques (using the "tangent residual" as a potential function) are novel. For the final version, the authors are encouraged to incorporate the reviewers' suggestions to improve the presentation.


**Award:**

Yes

---

### Decision · Program_Chairs · 2022-09-14

Accept